# Sexist Myths Emergency Healthcare Professionals and Factors Associated with the Detection of Intimate Partner Violence in Women

**DOI:** 10.3390/ijerph18115568

**Published:** 2021-05-23

**Authors:** Encarnación Martínez-García, Verónica Montiel-Mesa, Belén Esteban-Vilchez, Beatriz Bracero-Alemany, Adelina Martín-Salvador, María Gázquez-López, María Ángeles Pérez-Morente, María Adelaida Alvarez-Serrano

**Affiliations:** 1Guadix High Resolution Hospital, 18500 Granada, Spain; emartinez@ugr.es; 2Department of Nursing, Faculty of Health Sciences, University of Granada, 18016 Granada, Spain; 3Virgen de las Nieves University Hospital, Andalusian Health Service, 18014 Granada, Spain; vero_mm_1981@hotmail.com; 4San Cecilio Clinical Hospital, Andalusian Health Service, 18016 Granada, Spain; belen.esteban.sspa@juntadeandalucia.es; 5The Inmaculate Clinic, Andalusian Health Service, 18004 Granada, Spain; beabracero98@gmail.com; 6Department of Nursing, Faculty of Health Sciences, University of Granada, 52005 Melilla, Spain; 7Department of Nursing, Faculty of Health Sciences, University of Granada, 51001 Ceuta, Spain; mgazquez@ugr.es (M.G.-L.); adealvarez@ugr.es (M.A.A.-S.); 8Department of Nursing, Faculty of Health Sciences, University of Jaén, 23071 Jaén, Spain

**Keywords:** intimate partner violence, emergency department, readiness, healthcare professionals, multivariate analysis

## Abstract

This study analysed the capacity of emergency physicians and nurses working in the city of Granada (Spain) to respond to intimate partner violence (IPV) against women, and the mediating role of certain factors and opinions towards certain sexist myths in the detection of cases. This is a cross-sectional study employing the physician readiness to manage intimate partner violence survey (PREMIS) between October 2020 and January 2021, with 164 surveys analysed. Descriptive and analytical statistics were applied, designing three multivariate regression models by considering opinions about different sexist myths. Odds ratios and 95% confidence intervals (CIs) were considered for the detection of cases. In the past six months, 34.8% of professionals reported that they had identified some cases of IPV, particularly physicians (OR = 2.47, 95% CI = 1.14–5.16; OR = 2.65, 95% CI = 1.26–5.56). Those who did not express opinions towards sexist myths related to the understanding of the victim or the consideration of alcohol/drug abuse as the main causes of violence and showed a greater probability of detecting a case (NS) (OR = 1.26 and OR = 1.65, respectively). In order to confirm the indicia found, further research is required, although there tends to be a common opinion towards the certain sexual myth of emergency department professionals not having an influence on IPV against women.

## 1. Introduction

Intimate partner violence (IPV) is one of the most common forms of violence against women and includes physical, sexual, and emotional abuse and controlling behaviours by an intimate partner [1,2,3]. Globally, it is estimated that at least 30% of women who have had an intimate relationship have been victims of physical and/or sexual violence by their partner or former partner, psychological abuse being the most frequent type of violence [3,4,5]. Due to its magnitude, and because of the severe consequences for women’s and their children’s health, it is deemed a first-order public health issue [4,6,7,8,9]. Furthermore, although there are few data, reports from China, the United Kingdom, the United States, and Spain, indicate that the number of IPV cases reported, the number of victims requesting help, and the severity of injuries have increased since the outbreak of the COVID-19 pandemic [7,10,11]. The public health, economic, and social crisis derived from the current pandemic has led to women not resorting to health services until the final stages in the cycle of abuse, exposing them to a higher risk of IPV [10,12,13]. This reality has raised concern in several organisations, civil society representatives, and researchers, who have reasserted the need for effective intervention to prevent and fight such a phenomenon [14]. This is why identifying practices related to IPV screening and associated factors in different contexts is essential in order to develop practical and useful guidelines and principles applicable in the healthcare environment [15,16].

On the contrary, virtually every battered woman seeks public health services at some point in their lives and, compared to non-battered women, they usually do so more frequently [4,17,18,19,20]. Moreover, those who screen positive for IPV tests are more likely to experience some episode in the following months [15]. However, although healthcare professionals are acknowledged as essential pillars when addressing this serious problem [4,14], available evidence as to whether they should perform a universal screening for IPV applied to all women is controversial. O’Doherty et al., after a systematic review and a meta-analysis, concluded that there is not sufficient evidence to recommend it, since, although the identification of cases has moderately increased, with respect to actual prevalence, it does not improve referrals to support services or women’s health results [21,22]. Nevertheless, other authors are in favour of such screening since they understand that women who are victims/survivors are provided with more benefits than disadvantages, especially in the emergency department [15,23,24,25]. The World Health Organization (WHO) recommends that healthcare services should not apply a universal screening method or a routing enquiry method in order to identify women who have been abused. However, it does encourage healthcare providers to talk about this issue with every woman showing injuries or disorders that may be related to it [26].

To this effect, healthcare professionals should be alert and actively search for any suspicious behaviour, symptoms, or signs; moreover, women expect to be asked [15,27,28]. In fact, the perception of IPV victims/survivors regarding the attention received may be negative if abuse is minimised or not identified [29]. In this sense, if they are asked about abuse, they are more likely to talk about it [30,31], although evidence is consistent in showing the low detection reached in clinical environments [17,32,33]. Such rates fluctuate between 2% and 50% if asked on a routine basis, and between 45% and 85% if asked in the presence of injuries suggesting violence, depending, in turn, on the screening method employed [15,17].

Factors hindering or facilitating an appropriate approach to IPV cases are numerous and varied, being classified into the three main groups: related to organisational or institutional issues, to the victims, or to healthcare professionals [17,34]. Among healthcare providers, most studies are focused on identifying their attitudes, beliefs, and perceptions on detection and response to IPV. Insufficient training, a lack of skills in handling such cases, fear of legal implications, resistance due to own values or to the professional role performed, weak scientific evidence on the effectiveness of interventions, frustration, and not being able to stop abuse are signalled as some of the individual factors that have an influence on the approach to this problem [15,16,17,29,35].

With regard to opinions towards IPV, although they are usually positive, negative opinions have also been documented among healthcare professionals. Thinking that it is a private and personal problem related to women, which cannot be addressed by the public health system [16,36,37], or related to a specific group of women [38], the result of a personal decision [39], or that there may be reasons justifying it [40] have all been related to healthcare professionals’ decisions not to assess IPV, or to have a limited response [17]. These opinions are based on traditional beliefs or false myths about IPV, whose goal it is to justify, minimise, or deny this type of violence from men to women within a relationship [41,42]. By altering threat awareness in potential victims and/or awareness of guilt in batterers, IPV acceptance levels are increased, which may be translated into potential deficiencies in healthcare services (less implications) [41].

There are many other beliefs based on misogynistic myths about IPV. For instance, thinking that those responsible for the violence experienced by women are the women themselves for not breaking off such a relationship, or that if they are asked about it, they will be offended [16,43,44]. Other beliefs have been found, such as that the main reason lies in batterers’ alcohol or drug abuse [45,46,47], in the fact that they suffer from some kind of mental illness, or in social marginalisation or in a low sociocultural level [48]. However, evidence is usually consistent when showing that, although these factors may be a trigger, they do not cause IPV by themselves [49,50], rather, within the complexity entailed by such a phenomenon, they nourish the prevailing patriarchal system [46,48,51].

Emergency departments treat a large number of women, with it being estimated that between 6% and 54% experience IPV and/or will probably experience it by their current or former partners, especially physical abuse and sexual coercion and particularly if they have little social support [15,29,31,52,53]. It has even been reported that many women who are murdered by their partners/former partners had resorted to emergency services [31]. That is why they constitute a privileged attention unit, not only to treat any possible injuries, but also to actually help victims minimise their exposure to abuse, or even to get out of such situation [15,35]. However, the efficacy of emergency services in the detection of IPV cases, particularly during the current health crisis, and of the factors associated with detection by healthcare professionals, including opinions on sexist myths, requires further research [53,54]. Our hypothesis is that professionals showing favourable opinions on certain false beliefs or sexist myths perform a lower detection of IPV cases in emergency services. The purposes of this study were to analyse the capacity to detect IPV of healthcare professionals working in the emergency department in the capital city of Granada (Spain), and to identify the associated factors, and the mediating effect of opinions on certain sexist myths, in the detection of cases. 

## 2. Materials and Methods

### 2.1. Data Collection

This study is a cross-sectional study by means of a voluntary, anonymous survey self-completed in paper by healthcare professionals from three emergency departments of urban reference public health hospitals (two specialty centres and one trauma centre) in the capital city of Granada (Spain). Physicians and nurses were invited to participate, since they are usually the first people outside the family environment to whom women are directed and are the professional categories most likely to assist them in the course of their activity and, therefore, to detect IPV cases, thus forming a sufficiently homogeneous group to obtain consistent results, as previously done [35,55,56,57]. With a total number of 195 professionals, with 90% power and 5% accuracy, a sample size representing 128 participants was estimated. Data collection was carried out between October 2020 and January 2021.

### 2.2. Measurement Tool

The survey employed was the physician readiness to manage intimate partner violence survey (PREMIS), widely used at an international level and adapted to different populations [58,59], translated into and validated in Spanish [55]. The original instrument was created with the aim of assessing the education level, knowledge, opinions, and perceptions of the quality of performance of physicians with respect to IPV cases [60]. Opinions on IPV were configured through individual questions regarding attitudes and beliefs [60]. The Spanish version has shown appropriate internal validity, high reliability, and predictive capacity to assess the level of implication and participation in IPV cases by primary healthcare professionals [61], and which is now applied in the emergency environment. It is formed by 64 items divided into four sections: respondent profile, knowledge, opinions, and practices. There are eight factors in the opinion scale: opinion on team training, opinion on legal requirements, opinion on self-efficacy, opinion on alcohol and drug use, opinion on victims’ autonomy/understanding, opinion on the capacity to deal with IPV, opinion on barriers to IPV management, and opinion on the easiness related to the facilities in order to deal with IPV [55]. From the whole questionnaire, we selected 20 items from the following sections to reach the goals of this study, and whose 12 final variables are summarised in Table 1:
(a)Respondent profile: Gathers demographic and employment information and the type of training received on IPV of each participant (five items). Age and years of experience in the emergency department (quantitative) were later dichotomised, taking the median score as the cut-off point.(b)Knowledge: Enquires about the knowledge stated by healthcare professionals on IPV causes related to victims and batterers (two items). Later, a new variable was generated, real knowledge, by adding the correct answers to previous questions (gender/female + they use violence as a means of controlling their partners).(c)Opinions: We selected three opinion scales from the survey regarding women who experience IPV that are related to sexist myths: victim understanding (six items), alcohol/drug abuse (three items), and constraints perceived by healthcare professionals (two items) (Cronbach’s alpha = 0.72). The victim understanding scale was mainly constructed based on the opinions regarding women being responsible for not leaving a violent relationship and feeling offended if asked about IPV. In the case of the alcohol and drugs scale, it is focused on the idea that the use of such substances is the main cause of IPV. Eventually, the constraint scale considers that professionals do not have enough time to treat IPV, thus deriving responsibility to the institution. The items forming these scales are scored according to a Likert scale from 1 to 7, with 1 = strongly disagree and 7 = totally agree. Since some items are written in negative terms on purpose, they were inversely codified for their analysis, following the directions of the original survey’s creators [60]. Later, the scale scores were cut off, generating new dichotomous variables: unfavourable opinions (scores from 1 to 3) and favourable opinions (scores from 4 to 7) (three variables).(d)Practices: Assesses the detection of IPV cases by healthcare staff in the past six months and the screening type (two items).

A pilot test of the survey used in this study was carried out with 30 emergency department professionals (11 physicians and 19 nurses) who were requested to complete the survey (included in the final study) and to make voluntary qualitative comments on it. All participants indicated that the items were easily readable and understandable, although some questions were too specific to be applied in emergency services, but all of them considered them relevant. Exclusion criteria for participation in the research were not applied, since the intent was that it could reach everyone. The data were entered into an Excel database specifically designed for this study, and it was later exported to the Statistical Package for the Social Sciences (SPSS) programme, version 2, for Mac (IBM, New York, NY, USA).

### 2.3. Data Analysis

Descriptive statistics were applied to the entire sample, calculating percentages for qualitative variables and measures of central tendency for age and years of experience, and for opinion scales (quantitative). The opinion scale scores were analysed according to the professional category and participants’ gender, and the ANOVA test was employed for the comparison of medians on the basis of gender in each professional category, with values of *p* < 0.05 being deemed significant. Later, three multivariate logistic regression models were designed for the detection of IPV cases in the past six months, considering sociodemographic (sex, age, and professional category), employment, training, and real knowledge variables, and each dichotomised opinion scale (unfavourable opinions/favourable opinions). Thus, model 1 takes the victim understanding scale opinions into account; model 2, the alcohol/drug abuse scale; model 3, the constraint scale, together with the remaining variables under consideration. The adjusted odds ratios and confidence intervals are calculated at 95%, taking those who showed favourable opinions about said myths as a reference group.

### 2.4. Ethics Considerations

This study was authorised by all participating hospitals and complied with the good clinical practices set forth in Directive 2001/20/EC and Law 14/2007, of 3 July, on biomedical research. The personal data processing in health research followed the provisions of Organic Law 3/2018, of 5 December, on Personal Data Protection and Guarantee of Digital Rights. Informed consent from the professionals participating in this study was provided, and a favourable resolution of the Research Ethics Committee of the province of Granada under No 1044-N-20.

## 3. Results

In total, 180 professionals participated, which means a response rate of 92% of the professional personnel from the three emergency departments analysed. Finally, 16 surveys were dismissed for including over 20% incomplete items, so 164 were analysed. Sociodemographic and employment features, and those related to the training received by participants, are shown in Table 2. A total of 75% of the sample were women, most of them nurses (66.5%) with a median age of 41.48 years old (SD = 10.8) and with seven years of experience in the department (SD = 8.3). Out of them, 39.6% stated that they had read the healthcare protocol and 25% did not receive any kind of training on the topic at all.

As regards current knowledge, 42.1% of the professionals correctly identified that the main risk factor for becoming an IPV victim is the female gender, and 73.4% stated that it is generally true that batterers use violence as a means of controlling their partners. Both questions were answered correctly by 35.4% (real knowledge). When considering the implication in IPV cases, 65.2% of healthcare professionals stated that they did not detect any cases in the past six months. Among those who did detect some cases (*n* = 63), 54.9% stated that they enquired about patients who showed visible bodily injuries or indicators in their medical records, and only 1.2% asked every woman (Table 3).

Table 4 shows the quantitative scores for each opinion scale analysed by category and sex, and the percentages of professionals showing favourable opinions about sexist myths. The highest score was observed in the victim understanding and constraint scales (more in line with favourable opinions), both in physicians and in nurses, especially in male physicians (5.27, SD = 0.78) and female nurses (5.15, SD = 1.06). On the contrary, the lowest was found in alcohol and drugs as the cause of IPV, mainly in female physicians (3.76, SD = 0.82; *p* < 0.05). A smaller number of professionals agreed somehow with thinking that professionals do not have enough time to treat IPV victims (76.8%).

The results of the three multiple regression models for the detection of IPV cases are presented in Table 5. It is noted that those who show no favourable opinions to victims/survivors or with respect to alcohol/drugs abuse as causes of IPV, and the other variables remaining the same, are more likely to detect IPV cases at emergency services, although it does not reach statistical significance (aOR = 1.26 and aOR = 1.65, respectively). On the other hand, those who show more unfavourable opinions to time availability, reduce the identification (aOR = 0.78, NS). In every model, medicine professionals got more involved in IPV than nursery professionals. Very close to reaching significance, it is observed that having read the protocol increased detection of cases. In general, the fact of being a man, older than 40 years old, and having basic training, improved the result, while having real knowledge on IPV causes did not seem to do so (NS).

## 4. Discussion

The purpose of this study was to identify the factors associated with the detection of IPV cases among healthcare professionals working in the emergency departments of urban hospitals. Therefore, the results add to the research carried out in the last decades with similar goals [15], although we contribute to the assessment of opinions towards certain sexist myths.

The percentage of healthcare professionals who stated that they had detected IPV cases in the past six months (34.8%) is similar to the prevalence reported in our context. The last macrosurvey published in Spain recently, performed in a sample of 9568 women older than 16 years, reported that, among those who had or had had an intimate relationship with a man throughout their lives, 32.4% had experienced some type of violence [62]. Such a finding shows that the rates of implication by healthcare professionals seem to increase progressively, getting closer and closer to the actual prevalence [54]. However, although said figure is higher than the one reported in emergency departments by other researchers [15,24,29], it is still far from the estimated 54% of women who suffer IPV and attend an emergency department [15]. Therefore, there is a need to continue improving detection rates, although in the current context of the COVID-19 pandemic, we should take into consideration the influence of other related factors. It is reported that this pandemic has had a great psychological impact on Spanish healthcare professionals working in emergency departments, particularly in nursery staff [63]. The emotional consequences may reflect the uncertainty felt in the workplace in the presence of a possible outbreak of the infection, since they are on the front line of care [64]. Together with medical care pressure, it is very likely that changes in the priorities of healthcare provision have impacted problems other than COVID-19, such as IPV, which may have been relegated to second place, as it occurs with other severe pathologies [65]. On the contrary, this situation could offer some advantages to healthcare providers. For instance, the strong limitations on access to health centres could lead to overcoming one of the main constraints on the appropriate treatment of IPV victims/survivors, such as the fact that the person cannot visit the centre accompanied by their partner [24,66,67]. Professionals may take advantage of such a circumstance and achieve a closer approach, in a context in which the room for manoeuvre is quite reduced.

In this study, like in other investigations, the type of screening performed on a general basis by healthcare professionals was based on IPV objective indicators [16,24,29]. In Spain, emergency healthcare protocols in these cases encourage professionals to ask women showing specific signs or symptoms, in order to confirm violence [68]. Therefore, this practice follows the WHO’s recommendations [26], although it is widely documented that such actions do not contribute to identification [15,22]. In the United Kingdom, Canada, and Australia, it is also the usual practice [24,29,32], although in the United States, the recommendation is to perform IPV detection tests on every woman in their reproductive years, especially in emergency departments [69,70]. Hence, there is controversy over the guidelines to be followed, perhaps until evidence contributes more consistent results on the benefits for women, from whatever intervention.

Virtually all of the values of the scales considered were above 4 (above 7), allowing them to be considered moderate, in line with Short et al.’s findings [60]. However, the percentages of respondents who showed favourable opinions about every scale exceeded 75%. In relation to alcohol as the main cause of IPV, Ramsay et al. reported that 64% of their participants erroneously agreed with such a belief [47], which shows the high prevalence of such myths or stereotypes among healthcare professionals, particularly in our study’s participants.

Nevertheless, women showed lower scores on this scale, with statistically significant differences compared to men. Said results entail a novel finding that may reflect further knowledge about IPV aetiology by such professionals; however, this requires further confirmation.

The causal models designed in this study to determine the mediating role of favourable opinions about certain sexist myths, mainly those showing women or alcohol/drugs abuse responsible for IPV in case detection, contributed results that suggest said associations, regardless of the remaining variables. Noriega et al. found a direct correlation between sexist attitudes and a lower detection of IPV cases in a larger sample of professionals pertaining to different healthcare departments by employing an online survey with greater reliability than ours (Cronbach’s alpha = 0.85) [54]. Likewise, our indicia are consistent with the results from previous studies reporting said relationships [31,71]. On the contrary, having more time to treat IPV did not improve detection, which contradicts the opinion of numerous professionals, highlighting this constraint as one of the main causes not to intervene, especially in emergency departments [15,16,66,70]. Therefore, further research is required to clearly define the factors included in the model, and others that may be acting to change the plausible sense of this association.

Having received basic training on IPV and having read the protocol available suggest an increase in detection, as previously documented [54,61,72]. However, having real knowledge on the causes of IPV in our participants did not improve the result. That is why this study is in line with those that assert that a change of attitude (greater implications) does not depend exclusively on information or training, but rather, although they are necessary, they are not enough to face the complexity of such a phenomenon [17,54].

Regarding the participants’ sex, it is considered a significant predictor of the knowledge of and attitudes towards IPV, in which women would be better situated [56,57]. Moreover, it has been reported that victims may feel more comfortable about revealing their abuse experience to a healthcare professional who is the same sex [73,74]. Nevertheless, in our study, it was men who seemed to achieve higher detection rates. Such discrepancy between our and previous studies coincides with Ahmad et al., who stated the need for more solid and systematic research evaluating the impact of this variable on the implication towards IPV [15]. When considering age and years of experience, it is reasonable to think that professionals with maturity and expertise have a better understanding of IPV and greater skills to respond to victims/survivors of this type of violence [17,56]. Our indicia support these hypotheses, while there are studies that show contradictory results in this respect [58,75,76].

The variable associated, independently and consistently, with the detection of IPV cases in all models designed was the professional category. Medical professionals were more implicated than nursery professionals, a finding that disagrees with the results shown by Alvarez et al. in their revision of the literature in the primary healthcare services context [17]. Nurses have more and better opportunities to detect IPV, since they are the first healthcare providers and they spend more time with victims, as recognised by the International Council of Nurses [77]. However, although in every professional category a certain lack of training on the topic is detected [15,24,58,75,76], nurses generally believe that treating IPV is not part of their responsibilities [16,78], which may also be the opinion of those participating in this study.

## 5. Limitations

This study has several constraints. Although the internal validity may be considered high due to high participation, compared to those reported in other investigations of this kind [24,54,56,57,76], the external validity is limited. For generalisation of the results, participation of a higher number of health centres would be required, which should also be independent of the coverage area of our city. Regarding the design of the study, it was prepared on the basis of the constraints shown in other investigations with similar objectives, but which were not adjusted by relevant factors or did not consider detection as the result variable [56,58]. We proposed different explanatory models by introducing a series of independent variables supported by bibliography, and plausible mediating variables of detection such as opinions towards certain sexist myths. However, although the data were analysed as if it were an analytical study, the cross-sectional design only allowed us to suggest such associations among variables, and not a causality among them, so that results must be interpreted with caution.

With respect to the measurement instrument, we used a written, survey since the sample size was not very large and it was accessible in person in the very emergency department at an affordable cost, which improves participation. However, dependence on self-reports may introduce different types of biases in the results. On the one hand, the social desirability bias cannot be ignored, since we were dealing with a particularly sensitive topic in our context, namely, IPV. In order to minimise this bias, the anonymity of every participant was guaranteed. On the other hand, the unconscious bias of the provider to recognise abuse indicators and the memory bias may have influenced participants’ answers. In spite of this, our indicia agree with previous investigations reporting the influence of negative/sexist attitudes of healthcare professionals towards IPV on case detection, which supports the theoretical consistency of the study. In future investigations, the use of online surveys via e-mail should be proposed, both to attract a larger sample size in order to define clearly any possible associations observed and to improve results on sensitive topics [79].

Given the complexity of the explanatory frame regarding the relationship of opinions towards sexist myths in the implication of healthcare providers in IPV, other aspects not contemplated in this study would also have to be considered [15]. Noriega et al. pointed out that burnout in healthcare professionals is one of them [54], which, in the current pandemic context, has special relevance. Research with multilevel models and mixed methods would be a good choice to outline these and other factors involved in the detection of IPV cases in emergency departments by providing contextual information and information on said providers’ particular experiences.

Finally, in this study, IPV was only addressed with an orientation focused on women, although we are aware that men can also experience violence by their partner. However, sexist opinions, understood as misogynistic beliefs, an expression of patriarchy, also in the professional career context, are mainly directed at women, which is a structural, old, and universal issue [80,81,82,83,84].

## 6. Conclusions

Our results show that emergency department personnel in the city of Granada perform a moderate detection of IPV cases, with physicians, compared to nurses, getting more involved in the identification of this problem. These results bring to light the need, especially in nursing, to continue addressing gender-based violence by reinforcing professionals’ training to, among others, overcome the persistence of myths and false beliefs. Although we could not confirm the hypothesis formulated in this study, agreeing with certain sexist myths may suggest a significant role in decision-making when treating IPV, regardless of other factors, which requires further research. These findings could be used to propose intervention strategies, which, by reflecting the persistence of sexist myths, shall improve the attention offered to women exposed to IPV, together with training, the existence of detection and assistance protocols, or an adequate number of professionals, also from other fields.

## Figures and Tables

**Table 1 ijerph-18-05568-t001:** Definition of the variables derived from the Spanish version of the PREMIS survey.

Section	Items	Variables	Type of Variable
Respondent profile	Age in years, gender, professional category, and years worked in emergencies	Age (dichotomic)	Independent
		Gender: Male/female	Independent
		Professional category: Medicine/Nursery	Independent
	How much previous training about IPV issues have you had?	Time worked (dichotomic)	Independent
		Protocol read (no/yes)	Independent
		Basic training (≤20 h) (no/yes)	Independent
Knowledge	The strongest single factor for becoming a victim	Real knowledge: Gender/female + they use violence as a means of controlling their partners (no/yes)	Independent
	True statements about batterers		
Opinion scales	If victims of abuse remain in the relationship after repeated episodes of violence, they must accept responsibility for that violence	Victim understanding: Quantitative or qualitative (sexist/nonsexist attitudes)	Independent
	Victims of abuse could leave the relationship if they wanted to		
	If an IPV victim does not acknowledge the abuse, there is very little that I can do to help		
	If a patient refuses to discuss the abuse, staff can only treat the patient´s injuries		
	Healthcare providers have a responsibility to ask all patients about IPV		
	Screening for IPV is likely to offend those who are screened		
	Patients who abuse alcohol or other drugs are likely to have a history of IPV	Alcohol/drugs: Quantitative or qualitative (sexist/nonsexist attitudes)	Independent
	Alcohol abuse is a leading cause of IPV		
	Use of alcohol or other drugs is related to IPV victimisation		
	Healthcare providers do not have the time to assist patients in addressing IPV	Constraints: Quantitative or qualitative (sexist/nonsexist attitudes)	Independent
	I am too busy to participate in a multidisciplinary team that manages IPV cases		
Practice issues	In the past 6 months, which of the following actions did you take when you identified IPV: Did not identify IPV in past 6 months	Detection of IPV in the past 6 months (no/yes)	Dependent
	Check the situations listed in which you currently screen for IPV	Screening *	

* Descriptive analysis.

**Table 2 ijerph-18-05568-t002:** Sociodemographic and employment characteristics and training received by participating healthcare personnel (*N* = 164).

Variable	Percentage
Sex	
Man	25
Woman	75
Professional category	
Medicine	33.5
Nursery	66.5
Age in years	
Mean (SD)	41.48 (10.8)
(range) median	(22–63) 40
≤40	51.2
>40	48.8
Years of experience in emergencies (*n* = 160)	
Mean (SD)	7.27 (8.3)
(range) median	(0–37) 3.9
≤4 years	54.9
>4 years	42.7
Training	
None	25
Protocol read	39.6
Basic ≤20 h	28

**Table 3 ijerph-18-05568-t003:** Actual knowledge and practice issues about IPV in the past six months.

Items	Percentage
The strongest single risk factor for becoming a victim:	
Age (<30 years)	3
Partner abuses alcohol/drugs	42.7
Gender/female	42.1
Family history of abuse	36
Is generally true about batterers:	
They have trouble controlling their anger	32.9
They use violence as a means of controlling their partners	73.8
They are violent because they drink or use drugs	10.4
They pick fights with anyone	2.4
Real knowledge:	
Gender/female + they use violence as means of controlling their partners	35.4
Diagnoses of IPV you made in the past 6 months:	
None	65.2
Screening (among those who detected cases n = 63):	
All patients with abuse indicators on history or exam	54.9
Depressed/suicidal women	31.1
Every woman	1.2

**Table 4 ijerph-18-05568-t004:** Opinion scales by the professional category and sex.

Opinion Scales	Medicine			Nursing			Percentage of Professionals with Favourable Opinions
	Men	Women	*p*	Men	Women	*p*	
	*n* = 22	*n* = 33		*n* = 19	*n* = 90		
Victim understanding (Mean, SD)	5.27, 0.78	5, 1.20	0.352	5.04, 1.28	5.15, 1.06	0.701	96.3
Alcohol/Drugs (Mean, SD)	4.22, 0.85	3.76, 0.82	0.049	4.12, 1.03	4.13, 0.73	0.958	85.4
Constraints (Mean, SD)	5.27, 0.78	5, 1.20	0.352	5.04, 1.28	5.15, 1.06	0.701	76.8

SD: Standard deviation.

**Table 5 ijerph-18-05568-t005:** Multiple logistic regression models of each opinion to detect IPV cases.

	Model 1 Victim Understanding		Model 2 Alcohol/Drugs		Model 3 Constraints	
	aOR *	95%CI	aOR *	95%CI	aOR *	95%CI
Sex:						
Women	1	-	1	-	1	-
Man	1.19	0.53–2.61	1.21	0.54–2.69	1.17	0.53–2.60
Age:						
≤40 years old	1	-	1	-	1	-
> 40 years old	1.47	0.73–2.95	1.56	0.76–3.17	1.48	0.73–2.99
Professional category:						
Nursery	1	-	1	-	1	-
Medicine	2.58	1.24–5.38 **	2.47	1.19-5.16 **	2.65	1.26–5.56 **
Time worked:						
≤4 years	1	-	1	-	1	1
>4 years	0.90	0.16–5.15	0.94	0.16–5.36	1.25	0.21–7.15
Protocol read:						
No	1	-	1	-	1	-
Yes	1.92	0.95–3.89	1.98	0.97–4.03	1.91	0.95–3.86
Basic training <20 h:						
No	1	-	1	-	1	-
Yes	1.20	0.57–2.56	1.21	0.57–2.59	1.17	0.55–2.50
Real knowledge:						
No	1	-	1	-	1	-
Yes	0.54	0.26–1.15	0.53	0.2–1.12	0.53	0.25–1.13
Opinions:						
Favourable	1	-	1	-	1	-
Unfavourable	1.26	0.19–8.38	1.65	0.64–4.35	0.78	0.34–1.78

* aOR: Adjusted odds ratio; CI: Confidence interval; ** *p* < 0.05.

## Data Availability

The data that support the findings of this study are available from the corresponding author, upon reasonable request.

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
