# Peer review of "Sexist Myths Emergency Healthcare Professionals and Factors Associated with the Detection of Intimate Partner Violence in Women"

_ijerph, 2021, doi:10.3390/ijerph18115568_

Round 1
Reviewer 1 Report
The conclusion needs to be improved to provide a sound explanation of the findings: why the rate of detection is moderate and why physicians are more involved than nurses although nurses are better placed to detect IPV incidents? What kind of intervention strategies can be recommended based on the results? Any suggestions as to how to eliminate sexist attitudes?
The authors also need to copy-edit their manuscript.
Author Response
We welcome your comments and suggestions.
In the discussion section it is indicated that the indexes of the opinion scales are considered moderate, following the criteria presented by the creators of the PREMIS questionnaire. For its part, we consider the degree of case detection (34.8%) to be moderate, because although it is higher than that reported in other studies, not reaching 54%, which would be the estimated percentage of women victims of IPV who come to hospital emergency services according to the literature, we understand that it is not too far away.
Likewise, the discussion indicates the reasons why nurses can perform a lower detection of IPV cases, compared to medical professionals, according to the literature consulted.
Following your suggestions, the conclusions have been improved
Reviewer 2 Report
This is an important study since there is plenty of evidence that documents the negative health effects IPV have on victims.
I do have two questions:
I am not sure that the questions used to measure sexist attitudes really measure that. Rather, they seem to be myths surrounding IPV. If this study is going to be titled as a study on sexist attitudes towards IPV, a stronger link must be established that the questions used actually measure sexism. If that link cannot be established, then this really is a study of health care providers' endorsements of myths surrounding IPV.
Were any means testing on the Likert scale questions conducted? For instance, the difference between men and women, or nurses vs. doctors? This might give some more insight into the attitudes different genders or different health care providers have on the myths surrounding IPV.
Author Response
We welcome your comments and suggestions.
The title has been changed, and accordingly, the objective of the study to adjust to what the PREMIS questionnaire specifically measures, which are opinions based on certain sexist beliefs or myths.
We have incorporated the differences according to professional category and sex in the opinion scales analyzed in Table 4, as well as the result of the ANOVA test of comparison of means according to sex in each professional category, showing the corresponding p values.
Reviewer 3 Report
Strengths:
1. The current article seeks to describe how sexist attitudes toward victims of IPV influence the detection of IPV cases.
2. The authors utilize health care providers in the ED who are likely to treat patients with presentations related to IPV, but who also have significant barriers to identification of IPV.
Major growth areas:
- There is a short mention if how certain attitudes can be considered sexist (line 95-97); however, because the authors seek to describe the effect of sexist attitudes on IPV, it would aid the reader to define and discuss sexist attitudes in more depth, and how these opinions fit the definition of a sexist attitude, especially since most of the research literature has not defined these opinions in that way.
- There are no hypotheses stated.
- The justification of using both doctors and nurses in the ED is important to expand upon. Both professions have different training and training durations, and it is important to state in the methods that this will be used as a predictor of detection.
- Although the PREMIS measure is widely used and cited as reliable and valid, it is unclear how the measure was derived (empirically or theoretically). This is important because the knowledge subscale (in this study) is considered sexist/non-sexist and it is implied that these attitudes are either true or false given the statement in line 96. Are there data to suggest that these are indeed myths? Additionally, when we make changes to a subscale the reliability and validity of that measure changes and thus, an explanation for those changes is needed. Additionally, it is unclear how many opinion subscales there are and why the particular subscales were chosen. There needs to be some discussion of the decision points to change a measure or only use certain subscales.
- Although the authors assert that the PREMIS subscales can reflect sexist attitudes, this was not empirically tested by correlating the subscales with a validated scale measuring sexist attitudes. This is largely problematic because the whole study is framed as measuring sexist attitudes. If there is data that clearly connect the PREMIS subscales with a validated measure for sexism, then this needs to be explicitly stated and described.
- Table 2 does not seem consistent. The authors indicate that the median number of years of experience in the ED is 3.9, but only 5% of providers have experience that is less than or equal to 4 years.
- The discussion of the multiple logistic regression models was confusing. Did all models fail to predict detection of IPV? Additionally, the set up for Table 5 does not aid interpretation of the models and it is strongly recommended that there be some symbol to denote p > 0.05 if all or most of the findings were not significant and p < 05. Was provider gender, victim understanding, alcohol/drugs, and constraints used to predict IPV detection in one model and then age, victim understanding, alcohol/drugs, and constraints used to predict IPV in a second model. OR was each subscale used with the other covariates to predict IPV detection, if that is the case, an explanation for why all the PREMIS subscales were not entered into the same logistic regression model is warranted (such as multicollinearity).
- In line 112, the authors indicate that they were going to run a mediation analysis on the sexist attitudes subscale; however, it does not appear the authors conducted those analyses. It is important in the results section to report that a mediation analysis was not conducted as there were no significant associations among the subscales and IPV prediction.
- The discussion goes into depth on directions of associations, but if this reviewer understands correctly, there were no significant results in the logistic regression models.
Minor edits: line 47-49 wording is confusing;
line 64 systematic review instead of systematic revision.
Line 140 wording is confusing.
Line 173-174 it is unclear what the authors are trying to convey about who the reference group is.
Line 186: it is unclear what noncompliance reasons are
Author Response
Thank you very much for your comments. Please check the attachment.

Reviewer 4 Report
I recommend expanding the state of the question and presenting thetheoretical framework in which the study is framed.
Considering that this is a study that explores sexism in practice,
I am surprised that the presentation of the sample does not take into account
the gender distribution between medical and nursing staff.
It is not clear in the text that it is understood by sexist attitudes,
I think that the lack of theoretical framework contributes to this opacity. I consider that it makes no sense to interpret the results
without taking into account the socio-historical context
(patriarchy, pandemic situation, etc.).
I find that more depth is lacking in the analysis and in the discussion about the results obtained.
Author Response
We welcome your comments and suggestions.
Following their suggestions we have changed the title of the study to consider opinions towards certain sexist myths around IPV, rather than sexist attitudes, and accordingly, we have also changed the objective of the research. In the introduction, the theoretical framework of the approach to IPV in health services and specifically in hospital emergency services is presented, as well as the influence of the presence of sexist myths among health professionals and its influence on the detection of IPV cases. . Likewise, we have expanded the theoretical framework related to sexist myths and their relationship with the patriarchal system.
We have expanded Table 4 to indicate the distribution of the opinion scales considered taking into account the professional category and the sex of the study participants.
The first paragraph of the introduction refers to the current situation of the COVID-19 pandemic and the cases of IPV. The discussion raises possible consequences that the COVID-19 pandemic is having on hospital emergency services, as well as the opportunities that this situation can offer health professionals to improve the detection of IPV cases.
Likewise, the conclusions have been improved.